# Visual Grounding Helps Learn Word Meanings in Low-Data Regimes

## Abstract

Modern neural language models (LMs) are powerful tools for modeling human sentence production and comprehension, and their internal representations are remarkably well-aligned with representations of language in the human brain. But to achieve these results, LMs must be trained in distinctly un-human-like ways— requiring orders of magnitude more language data than children receive during development, and without any of the accompanying grounding in perception, action, or social behavior. Do models trained more naturalistically—with grounded supervision—exhibit more human-like language learning? We investigate this question in the context of *word learning*, a key sub-task in language acquisition. We train a diverse set of LM architectures, with and without auxiliary supervision from image captioning tasks, on datasets of varying scales. We then evaluate these models on a broad set of benchmarks characterizing models' learning of syntactic categories, lexical relations, semantic features, semantic similarity, and alignment with human neural representations. We find that visual supervision can indeed improve the efficiency of word learning. However, these improvements are limited: they are present almost exclusively in the low-data regime, and sometimes canceled out by the inclusion of rich distributional signals from text. The information conveyed by text and images is not redundant—we find that models mainly driven by visual information yield qualitatively different from those mainly driven by word co-occurrences. However, our results suggest that current multi-modal modeling approaches fail to effectively leverage visual information to build more human-like word representations from human-sized datasets.

## 1 Introductions

Neural language models (LMs) have achieved remarkable success across a wide variety of language processing tasks (Devlin et al., 2018; Liu et al., 2019; Radford et al., 2019; Brown et al., 2020). They have also proven useful for predicting aspects of human language processing, both in behavioral models of production (Arehalli & Linzen, 2020) and comprehension (Wilcox et al., 2020) as well as models of neural responses to linguistic inputs (Schrimpf et al., 2021; Caucheteux & King, 2022; Goldstein et al., 2022). LMs are therefore strong candidates as computational models of core aspects of human cognition. At present, however, these models are profoundly implausible as models of human cognitive *development*. The amount of training data required by effective LMs greatly exceeds the amount of linguistic input that human language learners receive during development (Zhang et al., 2020; Warstadt & Bowman, 2022): modern LMs are typically trained on tens of billions of sentences, whereas children only receive around a million sentences in the first three years of their lives (Bergelson & Aslin, 2017). These facts suggest that the mechanisms underlying language learning in LMs differ fundamentally from humans, and perhaps that inspiration from human langauge learning might improve the sample efficiency (and reliability) of LMs themselves.

One of the most significant differences between how humans and LMs learn language is that humans *ground* language in perceptual signals spanning many different modalities, including vision, touch, and hearing (Schroer & Yu, 2023; West & Iverson, 2017; Seidl et al., 2023; Clerkin et al., 2017). In sighted learners, vision is hypothesized to play a central role, as it delivers detailed information that is often directly coupled to linguistic input (Clerkin & Smith, 2022). Researchers in the natural language processing community have argued that multi-modal training might offer a path toward more human-like language learning (Bisk et al., 2020). Promisingly, recent years have seen the intro-

duction of a profusion of multi-modal models and learning algorithms, mostly targeted at tasks that require reasoning about data in both modalities simultaneously (Radford et al., 2021; Wang et al., 2022; Alayrac et al., 2022; Singh et al., 2022; Lu et al., 2022). However, the extent to which these models actually acquire more human-like representations of *language itself* has received limited attention.

In this paper, we investigate whether visual grounding can improve a key aspect of language understanding—*word learning*—in neural LMs. We study a variety of visually grounded models, including CLIP (Radford et al., 2021), GIT (Wang et al., 2022), and Flamingo (Alayrac et al., 2022), which represent drastically different ways of fusing vision and language data. While training these models, we carefully control, and systematically vary, both dataset size and the amount of within-language distributional information provided by word co-occurrence statistics. We then characterize these models with a suite of tasks designed to benchmark various facets of word learning, including syntactic categories, lexical relations, semantic features, semantic similarity, and alignment with human neural representations.

We find that grounded word learning indeed can yield better performance than the control language-only models in capturing word similarity and semantic features. However, this benefit is only observed when training on comparatively small-scale datasets. More surprisingly, it is observed only when limiting models' exposure to word co-occurrence information within language: in some cases, models actually exhibit *reduced* sample efficiency when learning from images and captions rather than images accompanied by single words. Although further analyses show that visual and distributional information are partially complementary to each other, none of the models we study can integrate both perceptual and textual contexts to yield improved word representations. Our results suggest that the learning of some, but not all, aspects of semantics can be facilitated by grounding, but that distributional information contained within language can override (and perhaps interfere with) visually grounded learning in current models.

## 2 BACKGROUND

**Word learning in children.** The present study investigates how visual grounding can help acquire knowledge of words and their meanings. Research has shown that children can correctly understand or produce words at a very young age, possibly even before the age of one (Frank et al., 2017; Bergelson & Swingley, 2012; Frank et al., 2021). Bergelson & Swingley (2012) measured children's attention to visual inputs when prompted with words and found that meanings of several common words are known by children from the age of 6 months onward. Using a different approach to measure word learning, Frank et al. (2017) collected the responses of parents to questionnaires about whether their children can correctly understand or produce words. Given the small amount of data required by children to exhibit these behaviors, word learning thus offers an appealing test-bed for comparative studies of LMs in a low-data regime.

**Multi-modality learning.** In recent years, multi-modality learning has seen significant advancements. For example, the CLIP (Radford et al., 2021) model is trained contrastively on 300M noisy (image, caption) pairs. It yields transferable visual representations as well as word representations that perform well on some tests of words similarity (Wolfe & Caliskan, 2022). GIT (Wang et al., 2022), by contrast, is a generative model, conditioning next-word predictions using visual inputs. It achieves state-of-the-art performance on multiple visual-language tasks, including image captioning and visual question answering. Diverging from both CLIP and GIT, as a final example, Flamingo (Alayrac et al., 2022) uses visual representations to modulate attention in a transformer language model, obtaining similar results. Since CLIP, GIT, and Flamingo all differ significantly from each other in how vision and language are fused, we test all of them in this work to explore what algorithm designs best benefit grounded language learning.

**Grounded and ungrounded learning algorithms as models of language acquisition.** Chang & Bergen (2022) investigate the word-acquisition trajectories in language-only models. However, their trajectories are computed only by measuring the change of model surprisal towards one word throughout training, which is less relevant to knowing word meanings. Huebner et al. (2021) and Warstadt et al. (2023) aim to improve language models trained on small datasets but focus more on grammar learning. Within grounded models, Wang et al. (2023) train image captioning models on first-person videos children receive (Sullivan et al., 2020) and claim that visual information

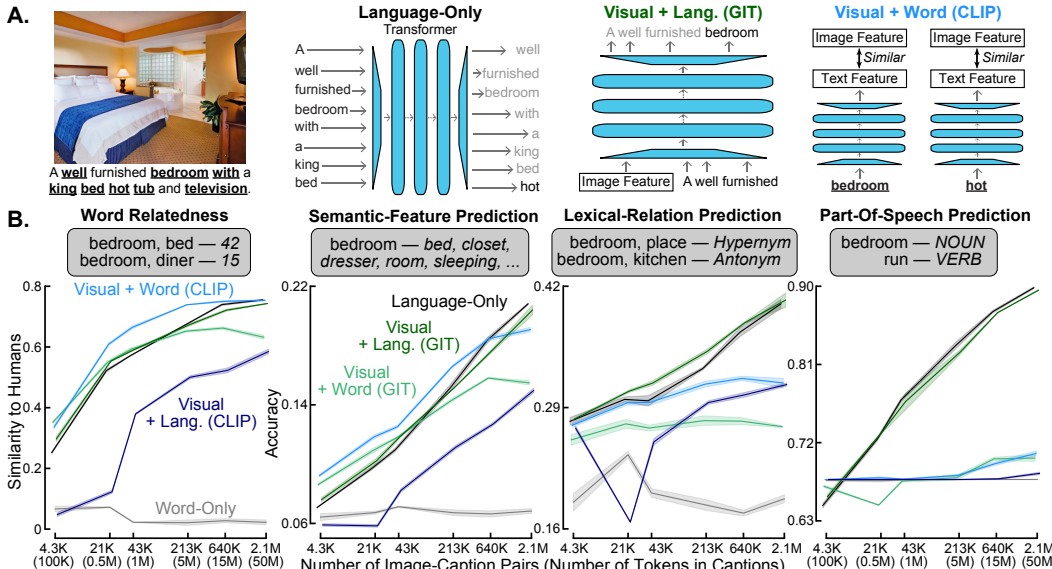

Figure 1: **In learning word meanings, visual information provides some help in low-data regime but only has limited additional utility relative to cross-word distributional information. A.** Pretraining schema for Language-Only, Visual + Word, and Visual + Language models. From left to right: an example image-caption pair, where only some of the words are separately used in Visual + Word models as labels; Language-Only models are trained on a next-token prediction objective; Visual + Language (GIT) models include the image features in the context to predict the next token; Visual + Word (CLIP) models optimize its text encoder to generate features that are similar to the corresponding image feature and dissimilar to other image features. **B.** Results on word-learning benchmarks. The word-relatedness benchmark computes correlations between hidden representations of two words and compares these correlations to human ratings of how related these words are. The other three benchmarks evaluate the accuracy of predicting the corresponding features of words (or word pairs) from the hidden representations of words (or the differences between word representations). The X-axis is in log scale. The width of lines in these plots represents the standard error of means across four models initialized from different random seeds.

helps predict words in context. Berger et al. (2022) and Portelance et al. (2023) also propose using multi-modality algorithms to understand word acquisition. Berger et al. (2022) specifically study acquisition of word categories, while Portelance et al. (2023) focus on learning function words from simplified visual stimuli. Our experiments aim at a significantly more general model of lexical knowledge, via a diverse collection of model architectures, more naturalistic visual supervision, and tests of numerous facets of word knowledge.[1]

## 3  METHODS

### 3.1  EVALUATION BENCHMARKS FOR WORD LEARNING

Comparing the learning efficiency between models and humans requires measuring their learning abilities on the same benchmark. However, it is challenging to directly ask models whether they can understand or produce words correctly, as is typically done in studies of child word learning (Fenson et al., 2007; Frank et al., 2017). Recording "attention" patterns to visual stimuli is also implausible for language-only models, which prevents a fair comparison between grounded and ungrounded models. We therefore evaluate LMs on tasks that measure the information content of their representations and the quality of their predictions.

---

[1]Although children clearly leverage other modalities to augment their language learning (Seidl et al., 2023), perceptual information is likely not necessary for learning word meanings. A large body of work shows that congenitally blind adults are still able to acquire "semantically-rich representations of visual words" (Campbell & Bergelson, 2022; Bedny et al., 2019; Minervino et al., 2018).

As described by Miller (1999), knowing the interrelations between words is critical to mastering different words. We use two benchmarks to characterize these interrelations: word similarity and lexical relation prediction benchmarks:

**Word similarity benchmark.** Word similarity benchmarks, such as WordSim-353 (Finkelstein et al., 2001), SimLex-999 (Hill et al., 2015), SimVerb-3500 (Gerz et al., 2016), and MEN (Bruni et al., 2012), assess how well models capture semantic similarities between a pair of words. We use the human judgments on word relatedness collected by Bruni et al. (2012) (see examples in Fig. 1**B**). For all benchmarks, we focus only on words typically acquired by children under the age of 10 (using information from Kuperman et al. (2012)), though we find this filtering barely changes the results. To extract similarity judgments from models, we extract word representations from a hidden layer, compute all pairwise cosine similarities between these word representations, and then compute Spearman correlations between model and human similarity judgments. The best layer is selected by the Spearman correlations for each model.

**Lexical relation prediction benchmark.** Lexical relation prediction benchmarks like CogALex-V (Santus et al., 2016a), ROOT09 (Santus et al., 2016b), and BLESS (Baroni & Lenci, 2011) focus on how accurate models can predict nuanced relations (synonym, hyponym, etc.) between words. We use the CogALex-V dataset (Santus et al., 2016a), which contains more than 2500 pairs of words in both training and test sets (see examples in Fig. 1**B**). This benchmark tests the accuracy of predicting the lexical-relation categories of two words using the difference of their hidden representations from models. These pairs are categorized into five lexical-relation categories: *synonymy, antonym, hypernymy, part-whole meronymy,* and *random*. For this and the following two benchmarks, the evaluation procedure then trains a linear probe on each layer to predict the targets from representations of words in a training set. The best layer is then selected by the accuracy on a validation set and its accuracy on a held-out test set is reported.

Understanding a word also entails more basic aspects of its meaning: such as the fact that *apples* are edible or *elephants* are large. We measure this with an additional benchmark:

**Semantic feature prediction benchmark.** In LMs, this can be assessed using semantic norm prediction tasks (McRae et al., 2005; Buchanan et al., 2019; Chronis et al., 2023). We use the dataset constructed by Buchanan et al. (2019), who ask human annotators to write down features of the word (see examples in Fig. 1**B**).

Beyond word-level representations, the "contexts in which a word can be used to express a particular meaning are a critical component of word knowledge" (Miller, 1999). Our experiments assess this knowledge via part-of-speech prediction and context-based word-understanding tasks:

**Part-Of-Speech prediction benchmark.** This benchmark evaluates the accuracy of predicting corpus-based (Davies, 2010) part-of-Speech (POS) tags for single words. Each word is labeled with its most frequent part of speech in the corpus (see examples in Fig. 1**B**). All words contained in the aforementioned three benchmarks are included.

**Context-based word-understanding benchmark.** This benchmark evaluates whether models can identify typical contexts in which words should appear. To create this benchmark, we first select real sentences containing each target word from `sentence.yourdictionary.com`, then minimally modify these sentences so that they are no longer appropriate environments for the target word. For example, if we have a target word *shoes* occurring in a sentence *Wear your shoes*, we might alter the containing sentence to read *Eat your shoes*, a significantly lower-probability environment for the target word. This is achieved by using a pretrained large LM (OPT-6.7B (Zhang et al., 2022)) to select the best modification from all possibilities to have the modified sentence less likely but still grammatical. By repeating this process, we obtain 280K sentence pairs for nouns, 128K for verbs, and 72K for adjectives, yielding three sub-benchmarks. To evaluate models on these benchmarks, we measure the fraction of sentence pairs in which they assign a higher probability to the original sentence compared to the modified one.

Another method to evaluate how well models represent context is to compare their representations to neural data. Thus, our final benchmark uses models' representations of sentences to predict the response of the language network in human brains:

**Brain-Response Prediction Benchmark** We use the brain response dataset collected by Pereira et al. (2018), which is also used by Schrimpf et al. (2021). We follow the fitting procedure proposed

by Kauf et al. (2023). A linear regression model is trained to predict the brain response to one sentence using its hidden representations. This model is then evaluated for its correlation between prediction and ground truth on the test set.

All words tested as targets in the first five benchmarks are designated as our words of interest. We show the results of pretrained LMs on these benchmarks in Appendix Fig. 6 and 7 as references.

## 3.2 MODEL TRAINING

**Dataset.** We sample image-caption pairs from the Conceptual-Captions-12M (Changpinyo et al., 2021) dataset (see Fig. 1**A**). Only images that were still valid as of August 2022 are used for training.

**Language-only models.** These models are trained on a next-token prediction objective using image captions alone, as in other neural language models (see Fig. 1**A**). In all experiments, we use a variant of the GPT-2 model architecture (Radford et al., 2019) with six layers. Other architectural parameters are the same as GPT-2 (see Appendix A.1). We also vary the number of layers and find it barely influences the results (see Appendix 8).

**Visual encoders and image features.** In visually grounded models (described below), we begin with pre-trained visual representations (trained without paired text–image data). These representations are taken from a Vision Transformer (ViT) (Dosovitskiy et al., 2020) pretrained on unlabeled ImageNet images using DINO (Caron et al., 2021), a state-of-the-art unsupervised learning algorithm. We use DINO-ViT also because earlier work (Zhuang et al., 2021; Konkle & Alvarez, 2022; Zhuang et al., 2022) showed that these unsupervised models share similarities with the human visual cortex. We use the representations of the [CLS] token at the last hidden layer as image features.

**Visual + language models (GIT).** GIT (Wang et al., 2022) treats the image features as part of the context and trains the models to predict the next words (see Fig. 1**A**).

**Visual + language models (CLIP).** CLIP (Radford et al., 2021) optimizes its text encoder to maximize the correlation between the representations of matching pairs (an image and its caption) while minimizing the correlation between non-matching pairs. In this study, we adopt the objective function proposed by Radford et al. (2021) to train language models, utilizing visual features precomputed from unsupervised visual networks. While we refer to these language models trained in this manner as "CLIP" models, it is important to note that they are distinct from the pretrained CLIP models developed by Radford et al. (2021). This distinction arises because the visual features are from the unsupervised pretrained models, the weights of the language module in our CLIP models are trained from scratch, and this module has fewer layers (six versus twelve).

**Visual + single-word models (CLIP).** To explore the isolated influence of visual information on word learning, we develop and test a single-word labeling method on images. These models, named Visual + Word models, are trained by first extracting all words of interest from one caption and then treating each of them separately as a label for the corresponding image (see the underlined and bold words in Fig. 1**A**). In Appendix Fig. 17, we show that removing this word-of-interest constraint does not change the performance. This single-word labeling method guarantees that word representations are learned exclusively from visual input, without incorporating (or receiving interference from) distributional information carried by co-occurring words. The same CLIP objective function is then used to train Visual + Word models (CLIP).

**Visual + Word Models (GIT).** Similarly, single-word labels and the GIT objective function are used to train these models.

**Word-Only Models.** These (trivial) models are optimized by predicting the single-word labels from just the [CLS] token inserted before the word. Here, the learning objective contains only information about word frequency, and no information about meaning or syntactic function.

**Training Details.** To explore how performance changes with respect to the dataset sizes, we vary the number of image-caption pairs from 4.3K to 2.1M (corresponding to 100K tokens to 50M tokens or 64K words to 32M words in captions). We train models for multiple epochs on these datasets, with training time determined independently for each dataset scale by the loss on the evaluation set (small datasets typically require more training than large datasets).

More details about training and evaluation can be found in Appendix A.1.

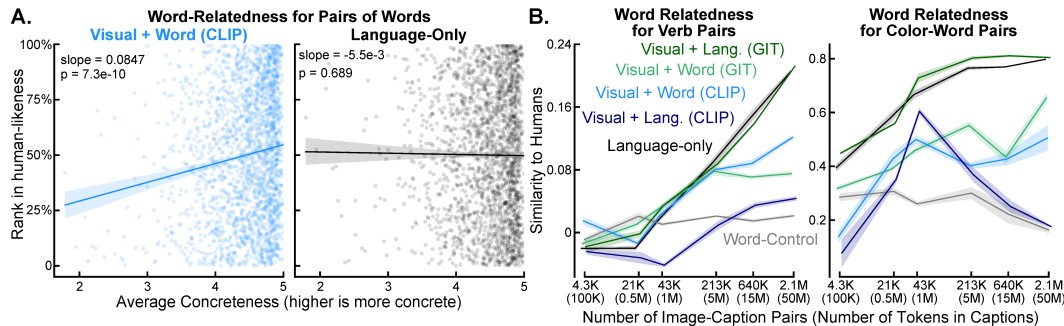

Figure 2: **Visual + Word and Language-Only models produce distinct representations. A.** Scatter plots for the word-relatedness benchmark. Each dot represents one pair of words. Its y-value represents the relative rank after sorting the word pairs using the difference between the human relatedness judgment and the correlation of model representations. A higher y-value means more human-like. Linear regression lines are plotted on the figure with the 95% confidence interval. **B.** The results of word-relatedness benchmarks on another dataset containing only verb words (SimVerb-3500, left) and the subset of color-word pairs in the previously used dataset.

## 4 RESULTS

### 4.1 VISUAL + WORD IS MORE EFFICIENT THAN LANGUAGE-ONLY ON LEARNING WORD MEANINGS, BUT ONLY ON SOME BENCHMARKS WITH SMALL DATASETS.

We first show the results of CLIP, GIT, and Language-Only models. These results show that when only a small amount of data is available, Visual + Word models are more efficient than Language-Only models in learning to relate words and predict semantic features (see the left two panels in Fig. 1**B**). CLIP achieves significantly higher efficiency in low-data regimes with single-word labels than GIT. However, Visual + Word models are worse than Language-Only models in identifying lexical relations between words and predicting POS tags of words (see the two right panels in Fig. 1**B**). In fact, Visual + Word models barely outperform Word-Only models in the POS benchmark. Even on the other two benchmarks where visual information delivers notable benefits, Language-Only models achieve comparable or better performance than Visual + Word models when learning from 50M tokens. To confirm that the results of Language-Only models are robust, we experimented with additional architectural and algorithmic variants and found similar or worse results (see Appendix Fig. 8). The lexical-relation results are also reproduced on two more datasets (see Appendix Fig. 9).

Given that visual information is useful for some aspects of word learning, we might then expect that visual and cross-word distributional information might be combined to yield even better learning efficiency. However, the typical way to combine them, implemented as Visual + Language models, fails to show benefits over Language-Only models. The Visual + Language (CLIP) models perform significantly worse than the Visual + Word (CLIP) models, indicating that the CLIP architecture is particularly inefficient in associating visual information to single words when full captions are present. As for the GIT architecture, although the Visual + Language (GIT) models are better than Visual + Word (GIT) models on most conditions, their performance trajectories seem to follow those of Language-Only models closely, and the improvement is minor. This limited benefit from combining visual and cross-word distributional information indicates that these two information sources compete with each other, and new learning mechanisms are needed to resolve this competition.

### 4.2 VISUAL INFORMATION HELPS LEARN CONCRETE WORDS.

The benchmark results illustrate that Visual + Word models diverge from Language-Only models; however, how this difference manifests per-word remains unclear. To better understand this difference, we first analyze three models with similarly strong performance on the word-relatedness benchmark: the Visual + Word (CLIP), the Visual + Language (GIT), and the Language-Only models trained with 2.1M image-caption pairs. By correlating model judgments with *each other*, rather than ground-truth predictions, we find that the Visual + Word (CLIP) and Language-Only models

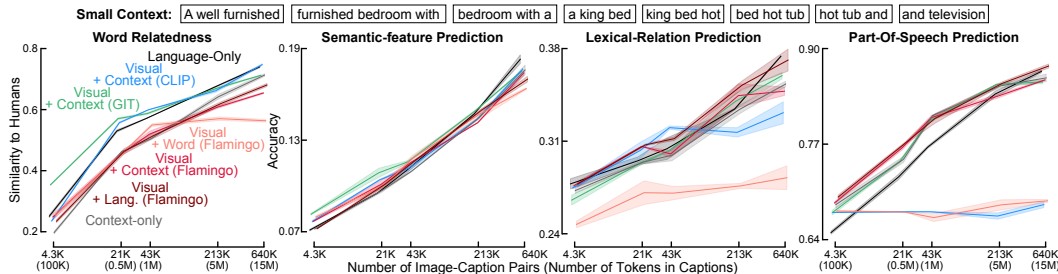

Figure 3: **Adding more context to Visual + Word models, or changing the model to Flamingo, offers little benefit.** The top row shows examples of small-context labels generated from the example caption in Fig. 1. Two models with different random seeds are trained in each condition.

significantly differ in how they relate words, while the Visual + Language (GIT) model yields almost the same judgments as the Language-Only model (see Appendix A.2 for details). This result shows that visual information can yield a different and useful representation. It suggests that GIT learns word representations predominantly from cross-word distributional information, not visual information, when both are available.

To further analyze differences at the word level, we then develop a human-likeness measure for a pair of words by comparing the judgments of humans and models (see Appendix A.2 for details). This measure is sorted to get a normalized "rank in human-likeness" measure for each pair of words, where a larger value means more human-like. Treating this measure as the dependent variable, we run linear regressions using different word features as independent variables (Tuckute et al., 2022): concreteness (Brysbaert et al., 2014), age of acquisition (Kuperman et al., 2012), Zipf lexical frequency Van Heuven et al. (2014), and prevalence Brysbaert et al. (2019). We find that the concreteness value best predicts the rank in human-likeness for the Visual + Word (CLIP) model, and clearly differentiates this model and the Language-Only model, as it is uncorrelated with the rank measure from the language-only model (see Fig. 2**A** and Appendix Fig. 10 to 12 for other features). This result shows one clear difference between Visual + Word and Language-Only models: grounded learning relates concrete words in a more human-like way than abstract words.

Since the dataset used in the word-relatedness benchmark contains mostly nouns and some adjectives (mostly color names) (Bruni et al., 2012), we extend this benchmark to other datasets to explore how the results change concerning word types. In another word similarity dataset focused exclusively on verbs (SimVerb-3500 (Gerz et al., 2016)), we find that Visual + Word models become significantly worse than Language-Only models (see the left panel of Fig. 2**B**). This is likely because image features from an unsupervised learning algorithm trained only on static images may contain very limited information for actions, which are better represented by dynamics in videos. In addition to verbs, we also find that the color words are differently related by visual models compared to human judgments (see the right panel of Fig. 2**B**). One potential explanation is that the human annotators view these color words as instances of a high-level word category, while Visual + Word models are over-dominated by the visual differences of color words. To confirm that these results are robust to the choice of word-relatedness datasets, we re-evaluate all models on SimLex-999 and find very similar results (see Appendix Fig. 13), meaning Visual + Word models are also better at relating nouns but worse on verbs than Language-Only models. Together, these additional word-relatedness results illustrate that learning only from static-image visual information cannot capture the full picture of human-relatedness judgments.

Finally, we extend our analysis to the semantic-feature prediction benchmark. As in the word-relatedness benchmark, we find that Visual + Word models predict the features of concrete words more accurately than Language-Only models (see Appendix Fig. 14). This means visual information helps to learn concrete words better than abstract words.

### 4.3 NARROWER TEXTUAL CONTEXTS IN GROUNDED MODELS YIELD LIMITED BENEFIT.

Visual + Word and Visual + Language represent two extreme conditions about how much distributional information is included. To explore the role of intermediate forms of distributional informa-

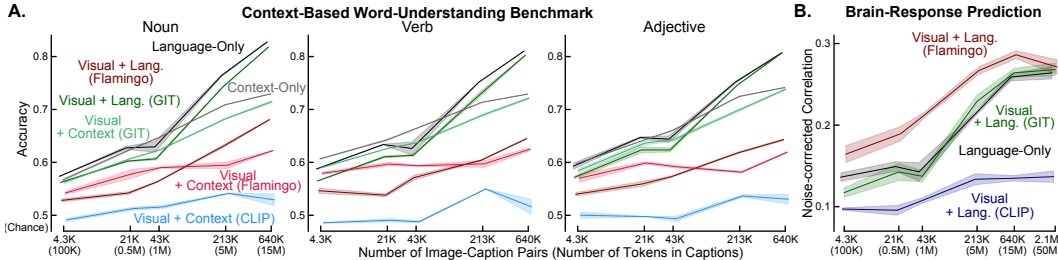

Figure 4: **Grounded models mostly underperform ungrounded models on context-based word-understanding and brain-response prediction benchmarks. A.** Performance of grounded and ungrounded models trained with small contexts or full captions as image labels on the context-based word-understanding benchmark. **B.** Brain-response fitting results for language-only and Visual + Language models. Four models with different random seeds are trained in each condition.

tion, we augment visual information with some, but not all, context around the words of interest (see the top part of Fig. 3). This leads to Visual + Context models and their ungrounded counterparts, Context-Only models. After testing these new models, we find that CLIP and GIT still fail to combine visual and distributional information to outperform Language-Only models. CLIP continues to illustrate a negative influence from incorporating more context (see Fig. 3). Visual + Context (GIT) models only outperform Context-Only models on small dataset scales. These results underscore the fact that new multi-modal models are needed to integrate both information sources effectively.

### 4.4 FLAMINGO ACHIEVES WORSE RESULTS THAN CLIP AND GIT.

To further evaluate the robustness of these results to choices of model architecture, we repeat a subset of our analysis in Flamingo models (Alayrac et al., 2022). This model architecture first extracts a summary vector from the image features and then modulates language representations using this vector through a cross-attention mechanism (see Appendix A.3 for details).

We find that these models underperform CLIP and GIT in leveraging visual information. On all benchmarks except the POS benchmark, Flamingo models perform no better than Language-Only models. Flamingo models also fail to benefit from more context than what is used in Visual + Context models. This might be why Visual + Language (Flamingo) models are more efficient learning in the POS benchmark, as Context-Only models also show similarly higher efficiency on this benchmark. To confirm that this result is robust, we vary an important hyperparameter in Flamingo but find it barely influences the performance (see Appendix Fig. 15).

### 4.5 VISUAL INFORMATION IS UNHELPFUL ON SENTENCE PROCESSING BENCHMARKS.

Because the four benchmarks used so far test word representations outside the context of sentence-level language understanding, we extend our evaluation to the context-based word understanding and brain-response prediction benchmarks discussed in Section 3.

All Visual + Language and Language-Only models are evaluated on the context-based word understanding benchmark. To understand how the local context of words influences this benchmark's performance, we also evaluate all Visual + Context models on it. Because these models are trained with at most three words, we split one sentence into multiple three-word segments whose center word is one of the words of interest. These segments are then sent to these Visual + Context models to compute their probabilities, where the Visual + Context (CLIP) models additionally receive an averaged image feature for the center word and use its embedding matching score as a proxy of the probability. The results of this benchmark show that grounded models underperform their ungrounded counterparts. CLIP and Flamingo show worse learning efficiency on this benchmark compared to Language-Only models. Only GIT models reach comparable results to their counterparts, likely because their word representations are almost fully determined by text (see Sec. 4.2).

As for the brain-response prediction benchmark, we only evaluate the Visual + Language and Language-Only models. As shown in Fig 4**B**, only CLIP shows significantly lower efficiency on

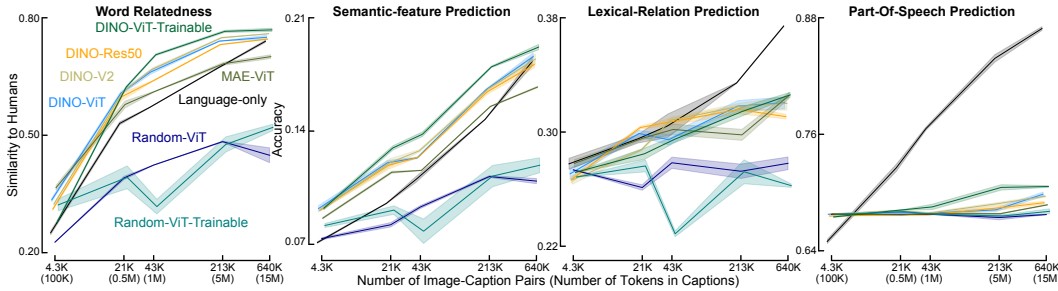

Figure 5: **Image representations have a small influence on word learning.** Visual + Word (CLIP) is used for these experiments. Two models with different seeds are trained in each condition.

this benchmark. All models produce higher prediction correlations when being trained on more data. Flamingo interestingly yields better results on small scales, which might be related to its higher performance on the POS benchmark. But all models except CLIP reach similar results when trained on 2.1M image-caption pairs. To summarize, these two sentence-level benchmarks show visual information is unhelpful for jointly understanding words and their context.

### 4.6 VARYING VISUAL ENCODERS YIELDS SMALL DIFFERENCES.

The visual information used in current models is computed from a ViT pretrained by DINO, and this ViT is not finetuned with language models. We first try finetuning the visual encoders and find that this significantly improves performance on word-relatedness and semantic-feature prediction benchmarks, though it saturates on larger datasets (see Fig. 5). This result indicates the potential of jointly training vision and language models. We then vary the pretraining algorithms with weights fixed after pretraining. Three more pretraining algorithms are tested, including a fundamentally different unsupervised learning algorithm (Masked Auto-Encoder (He et al., 2021)), an improved version of the same algorithm (DINO-V2 (Oquab et al., 2023)), and a random initialization (see Appendix A.4). We find that DINO-v2 yields very similar results, while MAE is significantly worse. The randomly initialized ViT yields non-trivial results but underperforms models with pretrained visual encoders. Changing the architecture of the visual encoder also barely influences the results (see DINO-trained ResNet-50). Although these results are from Visual + Word (CLIP) models, we find similar results on Visual + Word (GIT) models (see Appendix Fig. 16).

## 5 DISCUSSION

We have shown that language learning grounded in visual information helps current neural models acquire some aspects of word knowledge more efficiently than they can from text alone. Grounded models also learn qualitatively different representations from language-only models. But with reasonably large training data, ungrounded models become comparable to or even outperform grounded models. Even in low-data regimes, this benefit from visual information requires limited exposure to distributional information, as current learning algorithms struggle to integrate visual and distributional information.

One limitation of the present study is that the visual information used to augment language learning only contains representations of static images. Such images likely represent only a very small subset of all the visual information available to (sighted) language learners, who have access to the rich dynamics inherent in streams of visual inputs instead of independent images. Human language learners also perceive simultaneously from many other modalities like smell, taste, and touch. Finally, more work is needed to validate that these algorithms learn high-quality word representations from the same distribution of visual inputs that children receive (e.g. Sullivan et al., 2020).

Our results show that visual information can boost the learning of word meanings according to multiple measures. Future grounded LMs might thus serve as candidate models of how visual input can be leveraged to acquire language.

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

# A  METHODS

## A.1  TRAINING AND EVALUATING DETAILS

**Network Architecture and Tokenizer.** We use a six-layer Transformer network (Vaswani et al., 2017) for all models. The dimension for its per-token hidden representations is 768. There are 12 attention heads in each layer. The dimension of the intermediate layer in the post-attention feedforward layers is 3072. The tokenizer is from BERT (Devlin et al., 2018), whose vocabulary size is 30,522. The weights of the word-embedding layer are tied with the weights in the final output layer, which we find is critical to getting good performance for grounded models. As for the visual encoder, the features are extracted using the pretrained weights shared on Huggingface (Wolf et al., 2020), whose model ID is `facebook/dino-vitb16`.

**Optimization Details.** All models are then trained on these dataset for multiple epochs. The number of training epochs depends on the dataset sizes and is determined by monitoring loss values on the evaluation dataset. For all models except CLIP, we utilize a batch size of 128; CLIP is trained with a larger batch size of 512. We use AdamW (Loshchilov & Hutter, 2017) as the optimizer for training. The learning rate is linearly increased from 0 to 1e-4 over the initial 5000 steps, stabilizing at 1e-4 thereafter. The 100K-token models are trained for 200 epochs. The 500K-token models are trained for 40 epochs. The 1M-token models are trained for 60 epochs. The 5M-token models are trained for 20 epochs. The 15M-token and 50M-token models are trained for 10 epochs. These numbers are determined by observing how their validation loss changes.

**Word-Relatedness Benchmark.** We mainly use the human judgments on word relatedness collected by (Bruni et al., 2012), who asked annotators to judge whether one pair of words are more related than another pair of words. The authors selected words that are frequent both in online corpora including the British Web corpus (ukWaC) and as image tags (Bruni et al., 2012), yielding mostly concrete nouns. Each pair of words is compared to 50 other pairs that are randomly sampled. The relatedness of two words is quantified as the number of times this pair is judged as more related than the other pair in these 50 tests (see examples in Fig. 1**B**). Such relatedness scores are collected for 3000 pairs of words, of which 2057 pairs are used on this benchmark to focus only on words that are typically acquired by children under the age of 10 (Age of Acquisition is from Kuperman et al. (2012)). As for models, we use the cosine similarity between the hidden representations of two words from the same layer as the relatedness value for these two words. When the tested word contains more than one token, we use the representation from the last token. Finally, we compute the Spearman correlations between these similarity values from models and the relatedness scores from humans as results on this benchmark. For each model, we report the highest correlation across the results from all layers.

**Semantic-Feature Prediction Benchmark.** We use the dataset of psycholinguistic feature norms constructed by (Buchanan et al., 2019). The authors asked annotators to write down features of the word that they can think of. The response was then processed to generate single-word features (see examples in Fig. 1**B**), whose frequency of occurrences is used as the quantitative measure for the strength of this feature for one word. The original dataset contains 3,981 features and 4,436 words. These words undergo a further filtering process to retain only those with an Age of Acquisition (AoA) measure under 10, resulting in a final selection of 3,554 words. When testing models on this benchmark, we train a linear regressor to predict the feature strength of a word from its hidden representations. All the words are split into training (80%), validation (10%), and testing (10%) subsets and we generate two independent train-validation-test splits to reduce noise. Following the practice of Chronis et al. (2023), we train a partial least squares (PLS) regressor, where the number of components is set to be 100. We report the mean average precision (MAP) over the non-zero features of one word as the prediction accuracy. To compute this MAP, we first get the top-$k$ predicted features, where $k$ is the number of non-zero features in the ground truth, then count the number of overlapping features between the prediction and the ground truth, and finally normalize this count by

$k$. We determine which layer in the network to utilize for generating hidden representations based on the average accuracy observed in the validation set. The accuracy in the test set for this selected layer is reported as the evaluation result of one model on this benchmark.

**Lexical-Relation Prediction Benchmark.** The CogALex-V dataset (Santus et al., 2016a) includes 3,054 pairs of words in its training split and 4,260 pairs in the test split. The words whose AoA measures are higher than 10 are removed from the dataset, leaving 2704 training pairs and 3900 test pairs. The majority of the word pairs are in the *random* category (1944 of 2704 for training and 2770 of 3900 for test). To test models, we extract the hidden representations of two words and compute their difference as the representation for this pair of words. Following the practice of Ushio et al. (2021), we train a Multi-Layer-Perceptron network to predict the lexical relations. We use the default settings in the `MLPClassifier` class of `sklearn`, as we find varying these parameters only yields negligible performance difference. The macro average of F1 scores on the test set from the best layer is reported as the result of one model on this benchmark.

**Part-Of-Speech Prediction Benchmark.** The tags are generated by running Stanza (Qi et al., 2020) on the COCA-Fiction corpus (Davies, 2010), which contains around 100M words. For one word, we use the most frequent tag for it as its label on this benchmark (see examples in Fig. 1**B**). We include all words contained on the other three benchmarks and create four independent train-validation-test splits across words similarly in a 80-10-10 ratio. In each split, a linear Support Vector Classifier (LinearSVC) is trained. The average performance on all the validation sets is used to determine the best layer and the best hyperparameter (C) among (0.01, 1, 100). The test-set performance is reported as the result of one model on this benchmark.

**Context-Base Word-Understanding Benchmarks.** We select 140 nouns, 80 verbs, and 60 adjectives known to be acquired by young children (Frank et al., 2017), with 20 distractor nouns for each noun, 79 distractor verbs for each verb, and 59 distractor adjectives for each adjective. One pair of target and distractor words has 20 pairs of sentences. We download the example sentences for each target word from website `https://sentence.yourdictionary.com`. We then filter these sentences to only include sentences containing exactly one target word in its original form (singular for noun and present tense for verb). For each pair of target and distractor words, we sort all examples by the surprisal of the original sentence minus the surprisal of a changed sentence with the distractor replacing the target word (also called as the distractor-present-original sentence). This surprisal value is computed from the OPT-6.7B (Zhang et al., 2022) model. The smaller this difference is, the more reasonable this sentence is for this pair of words. We take the top 20 sentences with the smallest difference. For one sentence, we enumerate all possible replacements on all word positions except the target word. Each replacement yields one possible new sentence. A distractor-present-new sentence is further built from this new sentence by replacing the target word using the distractor word. The surprisal values for both the distractor-present-original sentence and the distractor-present-new sentence are computed from OPT-6.7B. A new sorting criterion is computed by $1.5 * S(dist_{new}) - S(dist)$, here $S(dist_{new})$ means the surprisal of the distractor-present-new sentence and $S(dist)$ means the surprisal of the distractor-present-original sentence. The new sentences from different ways of changing the original sentence are sorted by this new sorting criterion to get the sentence with the smallest criterion. This procedure is done to all 20 sentences to generate the 20 pairs of sentences for this target-distractor pair.

**Brain-Response Prediction Benchmark.** We use both two stimuli sets constructed by Pereira et al. (2018): one has 384 sentences split into 94 text passages and the other one has 243 sentences split into 72 text passages. For one stimulus set, the passages are randomly split into training (90%) and test (10%) subsets (Kauf et al., 2023). A linear regressor is trained on the training subset and evaluated on the test subset. This regressor uses the hidden representations of one layer of a model at the last token of the sentence to predict the corresponding voxel-wise brain responses. The performance of this regressor is computed as the Pearson correlation between the predicted and ground truth response, averaged across all voxels, and then normalized by the noise ceiling of this ground truth. This performance is further averaged across 10 splits and both stimulus sets to generate the benchmark result for one layer of a model. The best layer is selected based on its performance and generates the final score for this model.

## A.2 Analysis of Learned Representations

We compute the correlation between the judgments of different models, just like how these models are compared to humans. The correlation between the Visual + Word (CLIP) and the Language-Only models is only 0.70, which is significantly lower than their self-correlation values across different seeds (0.96 for the Language-Only model and 0.92 for the CLIP model). The Visual + Language (GIT) model yields almost the same word-relatedness judgments as the Language-Only model, as the correlation between these two models is 0.95, which is close to their self-correlation values (both 0.96).

The human-likeness measure for a pair of words between models and humans is defined as following. As the word-relatedness benchmark computes the Spearman correlation between a model and human judgments, we calculate two rank values for each pair of words, where one is from sorting all pairs by the model judgment and the other is from sorting all pairs by human judgments. The absolute difference between these two ranks is used to approximate how human-like one pair of words is represented by the model.

We also analyze whether this concreteness feature influences the performance of the semantic-feature prediction benchmark. The visual and language-only models trained with 640K image-caption pairs (15M tokens in captions) are used in this analysis, as their accuracy on this benchmark is very close. To compare these two models, we compute the accuracy difference between them for each word and find that the concreteness feature also positively correlates with this difference (see Appendix Fig. 14).

## A.3 Flamingo Training Details

The Flamingo architecture worked by having additional cross-attention layers modulating the outputs of text transformers, and these cross-attention layers were inserted at equal intervals of text transformer layers. The visual feature is processed by a Perceiver Resampler, whose number of layers is two and has 64 latents. Unlike GIT and CLIP models, the visual features sent to Flamingo models contain representations of all visual tokens. We test both inserting the cross-attention layers after every text transformer layer or after every two text transformer layers. The results of these two methods are in Appendix Fig. 15. We train the Perceiver Resampler, the cross-attention layers, and the text transformer layers from scratch using next-word prediction loss on image-caption pairs.

## A.4 Varying Visual Encoders

We use the MAE weights from Huggingface with model ID `facebook/vit-mae-base`. The outputs from the last hidden layer are averaged across all visual tokens to yield the image features. As for DINO-V2, we use the model `facebook/dinov2-base`.

# B Figures

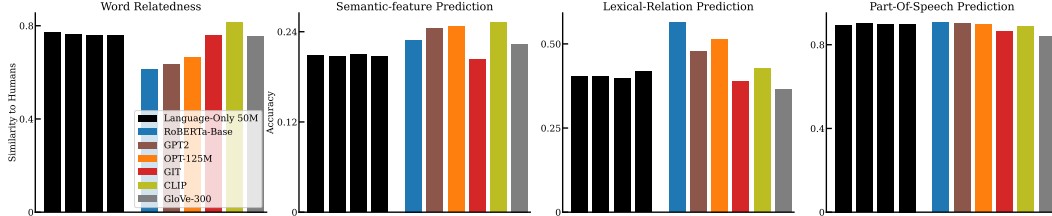

Figure 6: **Performance of pretrained models on word-learning benchmarks.** Bars with the same color represent models trained with different seeds. The Language-Only models shown here are trained on 50M tokens (2.1M image-caption pairs). Pretrained models are from huggingface. Their model ids are: "roberta-base", "gpt2", "facebook/opt-125m", "microsoft/git-base", and "openai/clip-vit-base-patch32". Note that only "microsoft/git-base" has the same architecture as the Language-Only models shown here.

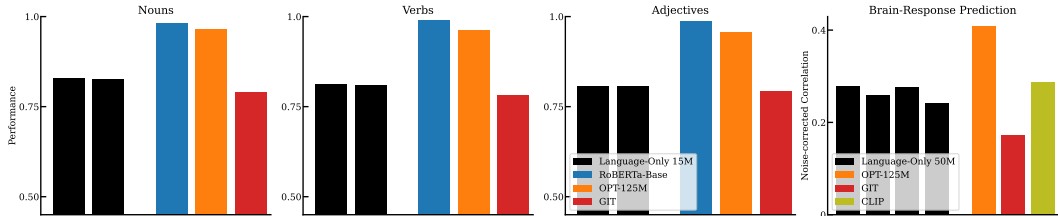

Figure 7: **Performance of pretrained models on context-relevant benchmarks.** Bars with the same color represent models trained with different seeds. The three left benchmarks are for context-based word-understanding benchmarks.

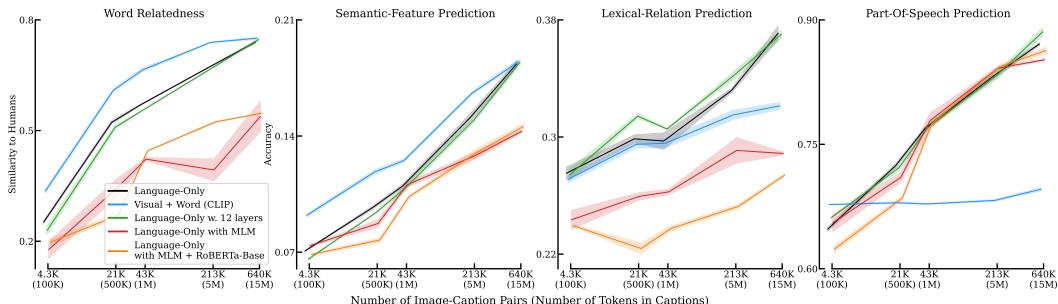

Figure 8: **Changing the network architecture or learning objective function in Language-Only models cannot improve the results.** MLM means Masked Language Modeling. Two models with different random seeds are trained in each condition.

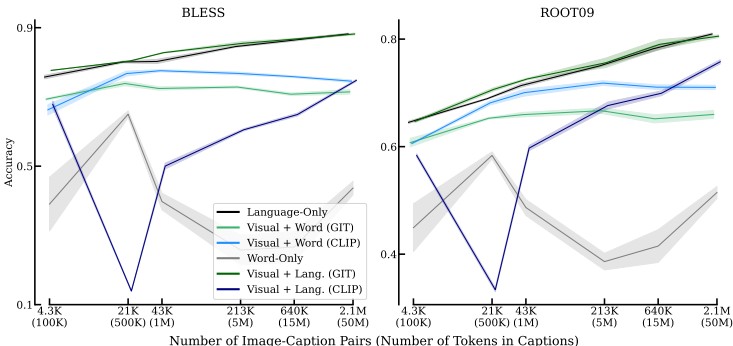

Figure 9: Results of lexical relation prediction benchmarks on two more datasets: BLESS and ROOT09. Four models with different random seeds are trained in each condition.

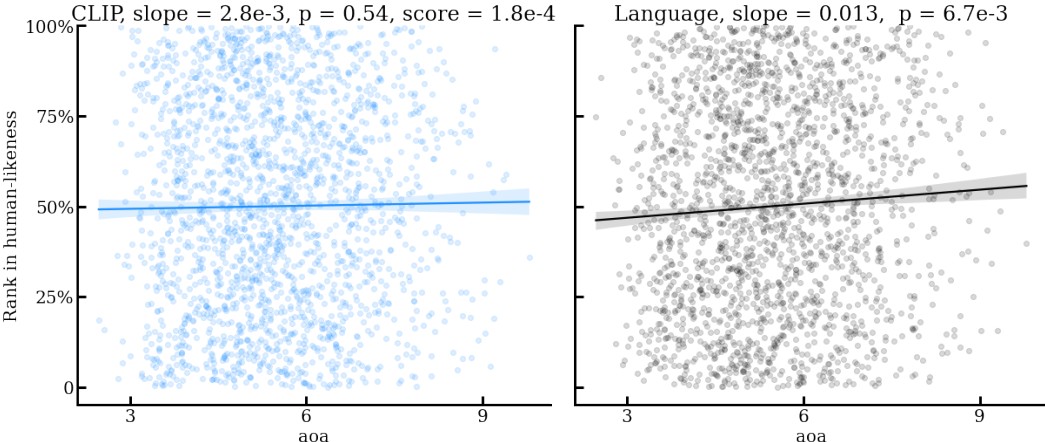

Figure 10: Scatter plots for the word-relatedness benchmark. The format is the same as Fig. 2**A**. Metric used here is AoA. The score means regression score (coefficient of determination), which is 0.018 for concreteness.

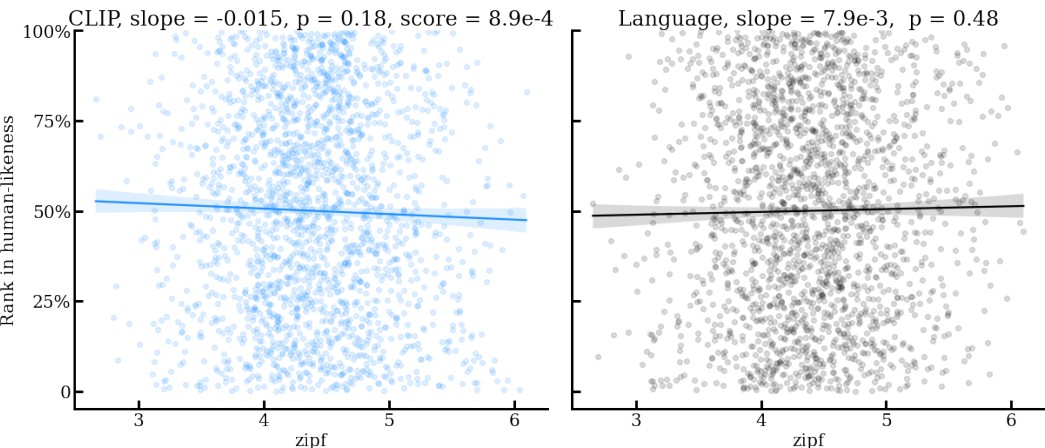

Figure 11: Scatter plots for the word-relatedness benchmark. The format is the same as Fig. 2**A**. Metric used here is Zipf.

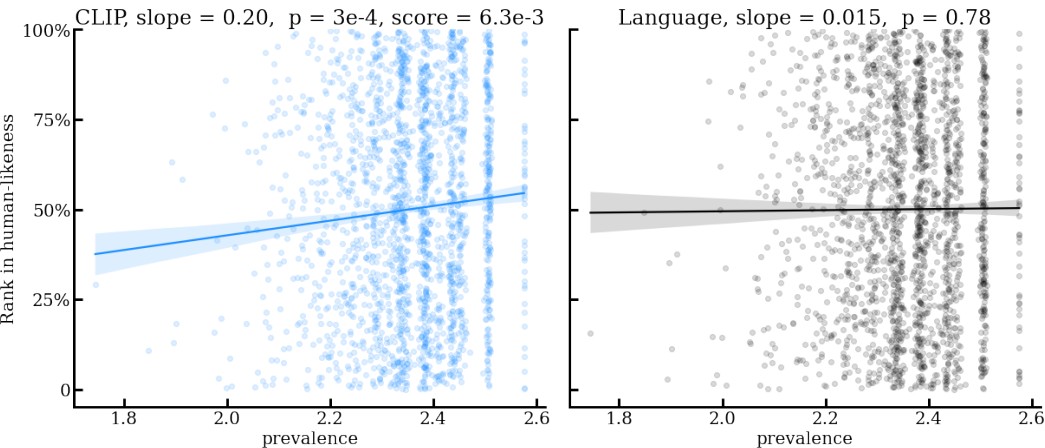

Figure 12: Scatter plots for the word-relatedness benchmark. The format is the same as Fig. 2**A**. Metric used here is Prevalence.

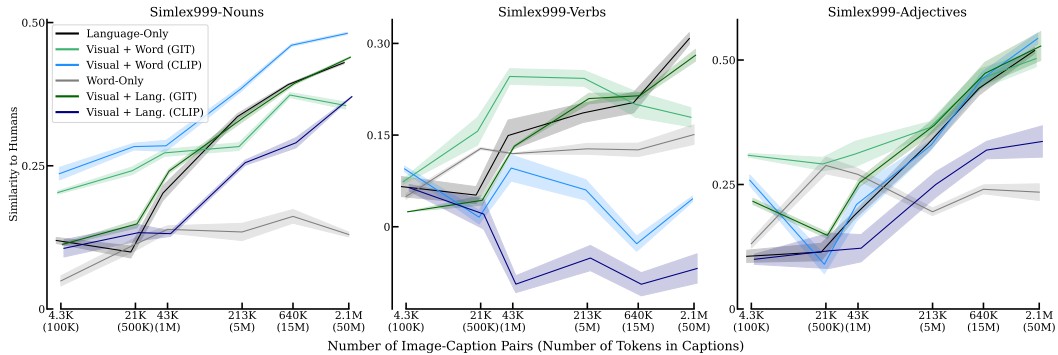

Figure 13: Results of word-relatedness benchmark on SimLex-999 (Hill et al., 2015) split by word categories.

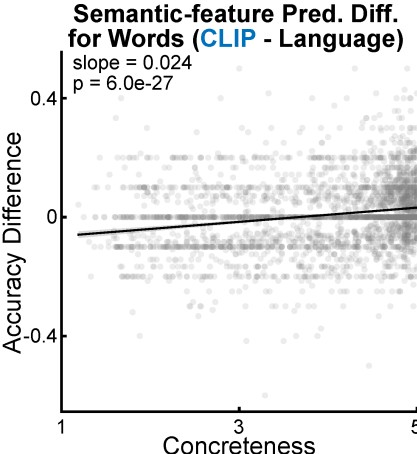

Figure 14: Relationship between the concreteness rating of one word and the accuracy difference between the CLIP and the language-only models.

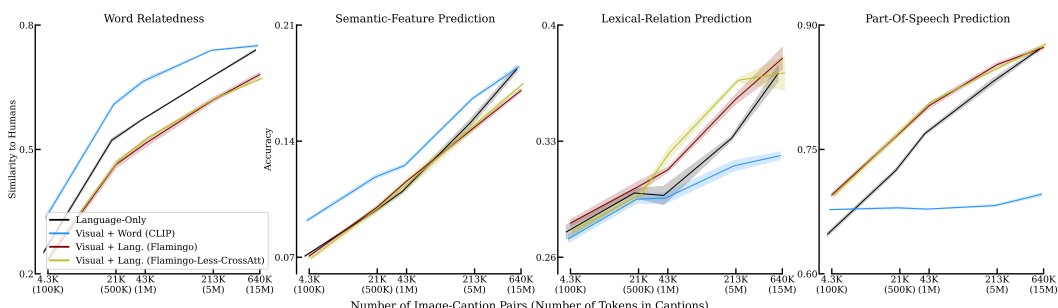

Figure 15: Reducing the number of cross-attention layers in Flamingo does not change its performance. Flamingo-Less-CrossAtt only has 3 cross-attention layers, while Flamingo used in the main paper has 6 cross-attention layers. Two models with different random seeds are trained in each condition.

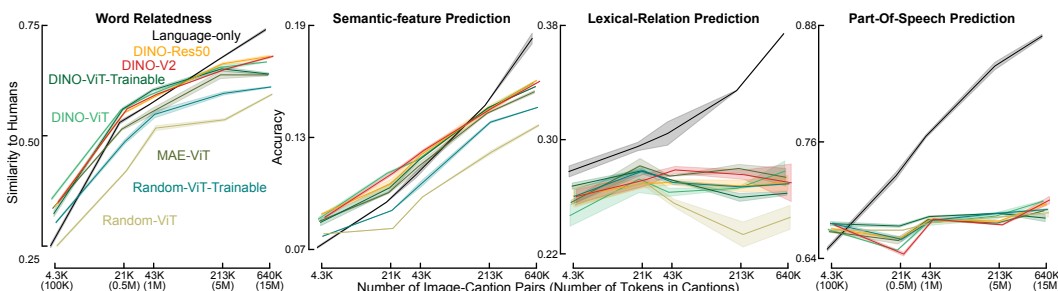

Figure 16: **Changing visual encoders leads to small influence on word-learning efficiency for Visual + Word model (GIT).** Two models with different random seeds are trained in each condition.

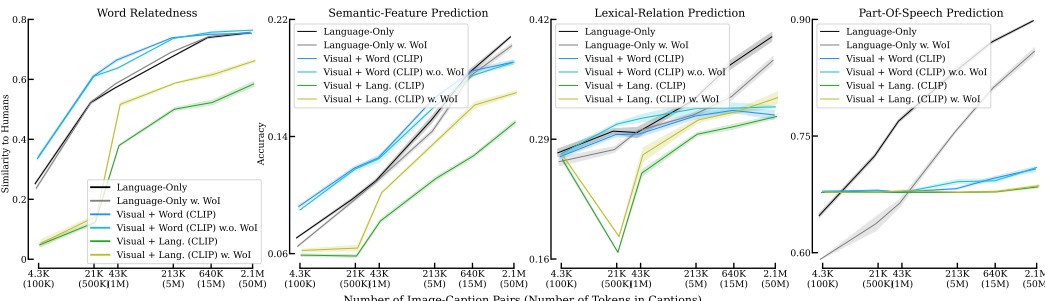

Figure 17: The word-of-interest constraint added to the Visual + Word (CLIP) models does not influence the results. Results of Visual + Language (CLIP) and Language-Only models trained with the word-of-interest constraint are also shown here, where the captions are rewritten to keep only the words that are tested in the word-learning benchmarks.

