# OpenReview forum: "Visual Grounding Helps Learn Word Meanings in Low-Data Regimes"
_ICLR.cc/2024/Conference — ICLR 2024 Conference Withdrawn Submission_

### Official Review · Reviewer_ooAw · 2023-10-14

**Soundness:** 2 fair
**Presentation:** 3 good
**Contribution:** 2 fair
**Rating:** 3
**Confidence:** 4

**Summary:**

The study investigates the effect of visual grounding upon language understanding of language models. For this objective, the authors perform the training of vision-language models in three settings, i.e. language-only, visual-language, and visual-word. Experiments show that visual grounding can aid language learning but mostly in the low-data regime.

**Strengths:**

The paper has the following strengths:

- The motivation is clearly stated and makes intuitive sense.

- The experiments are conducted thoroughly with various benchmarks, datasets, and prototype models.

- The obtained insights are strongly proven by the experiment results.

**Weaknesses:**

There are some details which can be raised from the paper:

- Even though the authors provide various evidence to substantiate their claim about the visual grounding ability to help language models, such claim is opposite from the findings attained by previous research of visual grounding [1,2,3] to certain extent. The paper lacks a discussion towards these research.

- Because language models haven proven their effectiveness in multiple applications nowadays, the contribution would become more appealing if the paper discusses what is the impact of its observation upon training language models.

[1] Retrieve, Caption, Generate: Visual Grounding for Enhancing Commonsense in Text Generation Models, AAAI 2022.

[2] Language Adaptive Weight Generation for Multi-task Visual Grounding, CVPR 2023.

[3] Visual Grounding in Video for Unsupervised Word Translation, CVPR 2020.

[4] Vokenization: Improving Language Understanding with Contextualized, Visual-Grounded Supervision, EMNLP 2020.

**Questions:**

- Why is the capacity of visual grounding for language understanding differently found from previous works?

- What benefit does the insight that visual grounding is beneficial for language understanding in low-data settings could provide for training language models?

---

> ### Author Response · Authors · 2023-11-14
> **Response**
>
> Thank you for your feedback. We appreciate the opportunity to address the concerns and questions you have raised.
>
> **How do our findings relate to previous work?**
>
> The question in this paper is whether visual information can improve the learning of linguistic representations. This is to eventually help reduce the amount of data that is needed to train the language models. The papers provided by the reviewer are all very different from the topic this paper investigates. All of them aim to improve the performance of certain tasks with the help of visual information. Specifically, [1] explores how visual grounding can help the task of generating common sense text from concepts. [2] focus on multi-modality tasks, exploring how language information can help referring expression comprehension and segmentation on images. [3] uses millions of videos and possibly billions of words to examine how word translation can be improved using video-caption joint training. [4] proposes a new basic architecture for vision-language joint training.
>
> Significantly, the studies cited in references [1,2,4] either rely on existing models pretrained on billions of words or, as in [3], require large multi-modal training datasets. In contrast, our paper assesses the performance of models trained on smaller datasets, using word learning benchmarks. Our goal is twofold: firstly, to investigate various factors affecting the learning efficiency of language models, and secondly, to deepen our understanding of human language acquisition processes through these models. These objectives distinctly set our work apart from the studies mentioned by the reviewer. Furthermore, our approach contributes to the ongoing efforts in training language models by demonstrating how reducing data requirements, inspired by language acquisition in children, can be effective.
>
> **What do our findings tell us?**
>
> Finally, our findings complement the methods proposed in these papers. To be more specific, we find that the visual information and the linguistic distributional information may compete with each other during learning. This finding suggests that carefully merging these two sources of information may lead to a better model, which can be added to the methods proposed in these papers. Moreover, these methods also provide new candidates to be tested on our evaluation benchmarks. Although they typically start from pretrained language models, we can adapt their methods to start from the Language-Only models trained only on the image captions and integrate the visual information afterward. We will explore this new class of methods in our future work.

---

> > ### Author Response · Authors · 2023-11-20
> > **Followup**
> >
> > We are writing to follow up on the earlier responses provided to your valuable feedback on our paper. As the discussion deadline is near, we would greatly appreciate any further comments or questions you might have. We're eager to clarify any remaining issues. Your insights are crucial for enhancing the quality of our work.

---

> > > ### Comment · Reviewer_ooAw · 2023-11-22
> > >
> > > Thank you for providing additional details of related work, which has clarified these problems for me.
> > >
> > > I have also read your rebuttal and the comments of other reviewers. I am quite aligned with Reviewer QLPJ's perspective. It appears that the current findings may not make significant contribution to the community. Moreover, the absence of practical applications discussed in the paper is a crucial concern. As a result, I preserve my initial evaluation score.

---

> > > > ### Author Response · Authors · 2023-11-23
> > > > **Response**
> > > >
> > > > We appreciate the response from the reviewer.
> > > >
> > > > **Title: What are the practical applications of our findings?**
> > > >
> > > > Current large-scale language models are typically trained on more than 200B words. The requirement of a huge amount of data during training significantly impedes further active explorations of new architectures and algorithms, not to mention its parallel need for extremely large-scale computational resources that are unaffordable to labs in universities. This is why we feel it’s critical for the community to also explore how to improve the learning efficiency of models, which is exactly the major research question we work on.
> > > >
> > > > To address this question, we take inspiration from how humans acquire language. Children can learn language from only millions of words. Their learning strategies differ from how models learn in multiple ways, including visual grounding. This paper proposes evaluations of several learning algorithms to explore the influence of these learning strategies on learning efficiency. Our work also made concrete suggestions on how better models should be created, including more organically merging visual and linguistic distributional information (see Sec. 4.2) and co-tuning visual and language representations (see Sec. 4.6). We believe this is an essential advancement toward achieving human-level efficiency in learning.
> > > >
> > > > **Title: What is our contribution to the community?**
> > > >
> > > > Our research adds to the approaches outlined in existing studies. Specifically, we've discovered an interaction where visual and linguistic distributional information may conflict during the learning process. This insight indicates that a more effective model could be developed by strategically combining these two types of information, enhancing the methodologies in these papers. The evaluation benchmarks constructed in this work, including the context-based word understanding benchmark, are also useful for the community.
> > > >
> > > > Our findings also shed light on whether, and how, visual inputs can shape language acquisition. The success of Visual + Word (CLIP) models in low-data regimes suggests a hypothesis: children might learn language by mapping visual inputs to single, loosely related words, similar to these Visual + Word models. These models' achievements are significant, especially since they map every word in a caption to an image, often in a noisy learning environment akin to what children experience. Despite this, they effectively grasp word meanings in small datasets and excel in some benchmarks. This indicates that children might use a similar single-word-visual-mapping approach in early learning. In future work, we aim to use children's learning patterns to further validate this hypothesis.

---

### Official Review · Reviewer_XApH · 2023-10-30

**Soundness:** 4 excellent
**Presentation:** 3 good
**Contribution:** 3 good
**Rating:** 6
**Confidence:** 4

**Summary:**

This paper investigates the influence of visual information on word meaning learning of language models.

The authors train CLIP, GIT, and GPT-2 models in a controlled way and examine the word learning results on various benchmarks such as word similarity and PoS tagging.

The experimental findings suggest that visual signals bring slight improvements to word learning, especially under low resource regimes. Interestingly, CLIP + Word predictions and language model predictions correlate poorly, indicating that contrastive learning captures a different pattern for word distribution. GIT performs similarly with pure language model, suggesting that it mainly learns from the cross-word distribution, struggling to balance between vision and text.

Further analysis shows that grounded learning relates to concrete words more like humans than abstract words. Additional results on vision encoder variants, Flamingo architecture, and sentence processing tasks further validate the main results of previous experiments.

**Strengths:**

- Investigating the effect of vision signals on language acquisition is an exciting topic, and the findings of this paper deepen our understanding of vision-aided language modeling.

- The experimental setup and results are comprehensive, with different types of models trained using the controlled network architecture and dataset, various benchmarks are adopted for evaluation and different angles are analyzed.

**Weaknesses:**

- The findings are weak in terms of practicality, as they cannot be directly translated into improvements for existing models.
- (Minor) Some experimental details are missing. See my questions (Q2 and Q3) below.
- (Minor) The results bullets are somewhat difficult to follow. I suggest the authors organize these findings into a more natural story to improve the reading experience.

**Questions:**

Q1: What are the implications of these findings for future research? I am not sure if the better concrete word modeling ability in low-resource regimes would be a bonus when we have abundant image-text pairs or single modality data to train models.  I think some previous explorations such as Vokenization [1], Initialization and plug-in fine-tuning [2] and Distillation [3] can be investigated (or discussed accordingly) with the findings in this paper.

Q2: How was the Flamingo model trained? Did you use the same 6-layer Transformer on the CC12M dataset? This seems not possible as it requires a special network for integrating vision information with cross-attention and an interleaved image-text dataset.

Q3: What representation of Visual + language models (CLIP) was used for word-based tasks? Did you extract the corresponding word representation from the full sentence to perform the downstream tasks?

[1] Vokenization: Improving language understanding with contextualized, visual-grounded supervision, Tan et al,  https://arxiv.org/abs/2010.06775

[2] How much can clip benefit vision-and-language tasks?, Shen et al,  https://arxiv.org/abs/2107.06383

[3] VIDLANKD: Improving Language Understanding via Video-Distilled Knowledge Transfer, Tang et al,  https://arxiv.org/abs/2107.02681

[4] Can Language Models Understand Physical Concepts? Li et al, https://arxiv.org/abs/2305.14057

---

> ### Author Response · Authors · 2023-11-14
> **Response**
>
> We thank the reviewer for the feedback and suggestions.
>
> **What is the practicality of our findings?**
>
> Our work explores how to improve the learning efficiency of models on small-sized datasets. This aims to reduce the amount of training data needed by the current large-scale language models and to better understand the human language acquisition process in children. Humans can learn languages from only millions of words. This indicates that inspiration from this acquisition process should facilitate the learning process of neural language models, which are currently trained with billions of words. Through exploring the influence of different factors, like visual grounding, on the learning efficiency of such models, our paper is also helping to reverse-engineer the language acquisition and cognitive development process in children.
>
> The reviewer also mentioned multiple previous explorations [1,2,3,4] Our findings can complement the methods proposed in these papers. To be more specific, we find that the visual information and the linguistic distributional information may compete with each other during learning. This finding suggests that carefully merging these two sources of information may lead to a better model, which can be added to the methods proposed in these papers. Moreover, these methods also provide new candidates to be tested on our evaluation benchmarks. Although they typically start from pretrained language models, we can adapt their methods to start from the Language-Only models trained only on the image captions and integrate the visual information afterward. We will explore this new class of methods in our future work.
>
> **More details about Flamingo training and CLIP evaluation**
>
> Like the other models, our Flamingo model used a 6-layer transformer as its language backbone. The Flamingo architecture worked by having additional cross-attention layers modulating the outputs of text transformers, and these cross-attention layers were inserted at equal intervals of text transformer layers. In our paper, we tried inserting the cross-attention layer after every text transformer layer or after every two text transformer layers. We found that the results of these two methods were similar (see Appendix Fig. 15). We have added more details in the paper (see A.3 in the Appendix). About the CLIP representations, we sent only the word to the model and extracted its hidden representations to be evaluated in our benchmark. We also tried to average the representations of the word across multiple sentences containing that word and found that the results were similar to the single-word representation result.

---

> > ### Author Response · Authors · 2023-11-20
> > **Followup**
> >
> > We are writing to follow up on the earlier responses provided to your valuable feedback on our paper. As the discussion deadline is near, we would greatly appreciate any further comments or questions you might have. We're eager to clarify any remaining issues. Your insights are crucial for enhancing the quality of our work.

---

> > > ### Comment · Reviewer_XApH · 2023-11-21
> > >
> > > Thank you for providing additional details of the evaluation setup, which has clarified these problems for me.
> > >
> > > Upon careful consideration of both your rebuttal and the comments from other reviewers, I find myself aligned with Reviewer QLPJ's perspective. It appears that the current findings presented in this paper may not significantly contribute to the community, particularly in light of previous related studies. Furthermore, the absence of practical applications stemming from these findings is a notable concern. Accordingly, I uphold my initial evaluation score.

---

> > > > ### Author Response · Authors · 2023-11-23
> > > > **Response**
> > > >
> > > > We appreciate the response from the reviewer.
> > > >
> > > > **Title: What are the practical applications of our findings?**
> > > >
> > > > Current large-scale language models are typically trained on more than 200B words. The requirement of a huge amount of data during training significantly impedes further active explorations of new architectures and algorithms, not to mention its parallel need for extremely large-scale computational resources that are unaffordable to labs in universities. This is why we feel it’s critical for the community to also explore how to improve the learning efficiency of models, which is exactly the major research question we work on.
> > > >
> > > > To address this question, we take inspiration from how humans acquire language. Children can learn language from only millions of words. Their learning strategies differ from how models learn in multiple ways, including visual grounding. This paper proposes evaluations of several learning algorithms to explore the influence of these learning strategies on learning efficiency. Our work also made concrete suggestions on how better models should be created, including more organically merging visual and linguistic distributional information (see Sec. 4.2) and co-tuning visual and language representations (see Sec. 4.6). We believe this is an essential advancement toward achieving human-level efficiency in learning.
> > > >
> > > > **Title: What is our contribution to the community?**
> > > >
> > > > Our research adds to the approaches outlined in existing studies. Specifically, we've discovered an interaction where visual and linguistic distributional information may conflict during the learning process. This insight indicates that a more effective model could be developed by strategically combining these two types of information, enhancing the methodologies in these papers. The evaluation benchmarks constructed in this work, including the context-based word understanding benchmark, are also useful for the community.
> > > >
> > > > Our findings also shed light on whether, and how, visual inputs can shape language acquisition. The success of Visual + Word (CLIP) models in low-data regimes suggests a hypothesis: children might learn language by mapping visual inputs to single, loosely related words, similar to these Visual + Word models. These models' achievements are significant, especially since they map every word in a caption to an image, often in a noisy learning environment akin to what children experience. Despite this, they effectively grasp word meanings in small datasets and excel in some benchmarks. This indicates that children might use a similar single-word-visual-mapping approach in early learning. In future work, we aim to use children's learning patterns to further validate this hypothesis.

---

### Official Review · Reviewer_ZHy9 · 2023-11-01

**Soundness:** 2 fair
**Presentation:** 3 good
**Contribution:** 3 good
**Rating:** 6
**Confidence:** 3

**Summary:**

This paper presents an analysis of the word learning / understanding capabilities of visually grounded models. The paper is motivated by human word/language learning which generally requires much smaller magnitudes of text data than language models, and thus investigates what differences lie in the word-learning dynamics/capabilities of models under different regimes of exposure to text and visual data (in the form of images) under a variety of evaluations pertaining to word learning.

Models evaluated have the following key properties:
* For language-only models a variant of GPT-2 is used.
* For visually grounded models, the two main families of models evaluated are CLIP and GIT, however Flamingo is also evaluated in one experiment.
* Models are also evaluated under varying sizes of text input windows, i.e. at the single-word level, at small contexts around target words, as well as at the full sentence level.

To assess word learning capability , a variety of evaluations are used:
* Word similarity -- how model similarity between words correlates with human similarity judgements.
* Lexical relation prediction -- training linear probes over model features to predict lexical relations such as synonymy or antonymy.
* Semantic feature prediction -- linear regression over model features to predict strengths of different features of words.
*  Part of speech prediction -- predicting part of speech tags from SVMs trained on top of model features.
* Context-based word-understanding benchmark -- a new benchmark presented with the paper, in which models are tasked with ascribing higher probability to correct contexts over perturbed distractors.
* Brain-response prediction -- linear regression over model features to predict brain response features to input text.

The paper finds that for a majority of experimental conditions, the multimodal models are generally either worse or comparable to the language-only model. Other findings include:
* A more fine-grained analysis in human ranking correlation controlled by word features (e.g. prevalence) finds that multimodal models perform better than language models on concrete words, which appears intuitive.
* Incorporating greater amounts of language context negatively impacts some multimodal model performance.
* Fine-tuning visual encoders improves performance.
* Models trained with smaller amounts of data benefit more from multimodality.

**Strengths:**

* The paper is compellingly motivated, and the research direction of understanding the word-learning capabilities of multimodal models is an important area that may help inform the development of future models.

* The paper presents an extensive set of experiments across a wide variety of experimental conditions for analyzing word-learning capabilities of multimodal models.

* The paper does a good job of summarizing its findings/conclusions gleaned from the experimental results, I found the summaries in the subsection titles and conclusions at the end of each subsection helpful in digesting the results.

**Weaknesses:**

* I believe this is a minor weakness, but I was left wondering what experimental results would be like across a wider variety of model families /variants. I was surprised to see the relatively lower performance of Flamingo, and wonder how models such as InstructBlip, LLaVA, and CM3 would perform.

**Questions:**

* For experiments evaluating similarities I'm wondering if any metrics other than the cosine similarity was used? I ask because as I understand it CLIP was trained explicitly to maximize the cosine similarity between matching image/text pairs, so it seems intuitive to me that its similarity judgement capability would be well evaluated under that metric. However, I'm not sure if this is necessarily true of other models, and I wonder if the observed trends might be any different under other metrics such as L2.

---

> ### Author Response · Authors · 2023-11-14
> **Response**
>
> We thank the reviewer for the positive feedback.
>
> **What experimental results would be for other models?**
>
> The reviewer is concerned about what results would be across a wider variety of model families or variants. Our selection of the diverse model architectures, including CLIP, GIT, and Flamingo, is supposed to improve the generalizability of our findings to other model families. That being said, the InstructBlip and LLaVa mentioned by the reviewer may belong to a different class of methods, and by the definition of their methods, large training datasets are necessary. This is because they start from pretrained large-scale language models already trained on billions of tokens, then finetune them on multi-modality tasks, and continue to finetune the resulting multi-modality networks using carefully selected visual-language instructions. The improvement of these methods over earlier ones is mainly on multi-modality tasks such as visual question answering and continuous chat with images, while we focus on evaluating the learning processes of linguistic information. It might be possible to replace the pretrained large-scale language models used in these methods with the Language-Only models in our paper and similarly do the multi-modality task finetuning and the instruction tuning, which could be a topic for future work. CM3 represents an even more different class of methods than InstructBlip and LLaVa, where retrieval on a large database is combined with generative models to do both text and image generation.
>
>
> **Why did Flamingo show worse performance?**
>
> As for why Flamingo worked worse than other methods, we have several hypotheses. The original Flamingo algorithm was designed to start from pretrained large-scale language models with the weights frozen and only train the cross-attention modules. The Flamingo authors even reported lower performance when the weights of language transformer layers were also fine-tuned. Therefore, one hypothesis is that this algorithm cannot efficiently train a neural language model from scratch with the cross-attention modules. Another hypothesis is that this algorithm may work less well with the image-caption pair data used in our work than the large mix of image-caption pairs, website texts embedded with multiple images, and captioned videos used in the original paper.
>
>
> **Metrics other than cosine similarity on the word-relatedness benchmark**
>
> The reviewer also asked whether metrics other than cosine similarity were used in the word-relatedness benchmark and mentioned that this metric could be easier for CLIP models to do well than other metrics. After testing the L2 distance metric suggested by the reviewer, we indeed find that CLIP models perform worse under this metric in the word-relatedness benchmark than the same models tested using cosine metric, while the performance of Language-Only and GIT models is not influenced by this change of metric. Since cosine similarity is closely related to the L2 distance metric with both vectors L2-normalized, this result suggests metrics influenced by the L2-norms are under-leveraging the capacities of CLIP models. This fits the reviewer’s expectation and is likely because the L2-norms of the hidden representations in CLIP are not optimized by the cosine-similarity objective function during learning and, therefore, do not carry much meaningful information. However, we believe this new finding should be interpreted as support for using the cosine similarity metric in the word-relatedness benchmark since this metric is more robust to model classes and yields results that are no worse than other metrics.
>
> Moreover, the results in the word-relatedness benchmark are not the only results supporting our claim. The semantic-feature prediction benchmark, for example, also shows evidence of benefits from visual information in low-data regimes, and this benchmark is evaluated through training regressors directly on the hidden representations.

---

> > ### Author Response · Authors · 2023-11-20
> > **Followup**
> >
> > We are writing to follow up on the earlier responses provided to your valuable feedback on our paper. As the discussion deadline is near, we would greatly appreciate any further comments or questions you might have. We're eager to clarify any remaining issues. Your insights are crucial for enhancing the quality of our work.

---

> > ### Comment · Reviewer_ZHy9 · 2023-11-23
> >
> > Thank you to the authors for a thoughtful response and for clearing up my questions.
> >
> > After consideration of the other reviews and discussion I echo other reviewers in agreeing with reviewer QLPJ's assessments. Although I believe the discussion around human language learning is a good motivator for the experiments presented in the paper, I agree with reviewer QLPJ and am of the opinion that asserting any statements/claims about human language learning itself as a conclusion from these experiments is too strong.
> >
> > With that said, I retain the belief that the learning efficiency findings may be of interest to members of the community working with the types of models evaluated in the paper.
> >
> > Therefore, I have updated my score to 6.

---

### Official Review · Reviewer_QLPJ · 2023-11-05

**Soundness:** 2 fair
**Presentation:** 2 fair
**Contribution:** 2 fair
**Rating:** 5
**Confidence:** 4

**Summary:**

This paper investigates the problem of visually grounded word learning, and arrives at the main conclusion that visual grounding mainly help acquire word meanings in low-data regimes. The performance of word acquisition are measured by calculating the similarity between model prediction and human annotations. Other side findings are also presented in the paper.

**Strengths:**

- Comprehensive evaluation on a wide range of tasks.
- While some figures are hard to read, the paper is generally well-written.

**Weaknesses:**

- Motivation of the work: the main metric is the similarity between model prediction and human judgments, but there are at least two steps that may result in information loss, and I'm not sure how reliable the conclusions are with the presented approaches:
    1. The authors used the cosine similarity between word representations to measure how the models acquire words, but other useful information that affects model preference may be encoded in the deeper architectures.
    2. There may be disagreement between human annotators. For example, Brysbaert et al. (2014) have the word *can* labelled highly concrete, which can cause quite high disagreement among people.
- Plausibility in terms of data exposure: while [*these (text-only) models are profoundly implausible as models of human cognitive development*] (Page 1), isn't the finetuning CLIP approach similarly implausible? Humans do not pre-train their visual and textual understanding systems on large parallel data; instead, human acquisition of words arguably happens in an incremental way.
- Arbitrary definition of *human-likeliness* (Appendix A1.2). The evaluation metrics, while reflecting the model acquisition of word meanings to some extent and from certain perspectives, do not necessarily reflect the model's ability to learn words.
- The line plots with multiple lines (Figs. 1B, 2B) are largely imperceptible. It would be better to use different line styles.

**Questions:**

- For CLIP-based settings, did you use a pretrained CLIP model or use the model architecture/objective with random initialization to train from scratch? If the former, isn't it exposed to many more image-caption pairs than your training data? If the latter, I'd be surprised that 4.3K pairs can lead to a decent performance and would meanwhile suggest the authors rename their models---CLIP is usually used to refer to the pretrained CLIP model.
- Why did you specifically pick CLIP, Flamingo and GIT as the model? There are several models, such as [Kiros et al. (2014)](https://arxiv.org/abs/1411.2539) and its variants, working on learning visually grounded word and sentence representations. In terms of performance on image-caption retrieval or generation, they might not be competitive with recent work, but the task of this paper is to investigate the acquisition of word meanings, and there's no reason to stick to the recent popular models.
- (minor) In the first sentence, you probably wish to use *NLM* instead of *LLM* as the abbreviation of *neural language models*?

--------------------updates after response--------------------------------------------------------

My apologies for not realizing the fact that the authors cannot see reviewers' comments after the rebuttal period, so I'll just post it here. I can see the rationales of the authors. All of them are, to some extent but not completely, convincing to me. The additional experiments are interesting, but I'm not sure if the authors have controlled carefully to ensure the fair comparison (sorry I should've realized and brought this up earlier) between visual + word and visual + context: if both of them are trained for the same number of epochs, visual + word will see #(words) - #(sentences) times more images than visual + context, while seeing the same amount of text.
I also have similar feelings on the child language acquisition claims---yes, the visual + word learning protocol is a reasonable hypothesis of human language acquisition, and this paper presents some evidence to support the hypothesis, but there is still something arguable in terms of experiment settings, and how strong the evidence is.

I've raised my rating to 5 to show my appreciation of the authors' response; however, I still think this paper needs some substantial work before publication.

---

> ### Author Response · Authors · 2023-11-14
> **Response**
>
> Thank you for your feedback. We appreciate the opportunity to address the concerns and questions you have raised.
>
> **Why these evaluation metrics?**
>
> First, we emphasize that our study does not solely rely on the cosine similarity between word representations. This metric is only used in the word-relatedness benchmark. The other three benchmarks shown in Fig. 1 do not look at similarity, but instead encodings of syntactic and semantic information in embeddings, as well as their usefulness in language modeling tasks. These additional benchmarks are designed to capture various aspects of word acquisition and understanding, thereby providing a more holistic view of the models' capabilities.
>
> We also wish to emphasize that many of our evaluations use deep representations, not representations from the word embedding matrix. As discussed in Section 3.1, we optimize the layer from which representations are drawn separately in each model. For CLIP models, later layers are better than earlier layers in general. For GIT models, earlier layers are typically the best. Moreover, the two benchmarks shown in Fig 3 test context representations of models, including surprisals and hidden representations of whole sentences. Both of these benchmarks evaluate more than just word representations.
>
> The reviewer also mentioned that there could be disagreement between human annotators. This is a good point, but we don’t think it influences the conclusions presented in this paper. The inconsistency between human annotators leads to a ceiling of the performance on the benchmarks. However, when two models are compared on the same benchmark, a higher number still indicates that the word representation is more human-like. Since our conclusions in this paper are drawn from comparisons of models, they should still hold even if human annotators disagree to some extent.
>
> Finally, the specific definition of the “human-likeness” metric used in A1.2 relates only to correlation with human judgments on the word-relatedness benchmark. Our paper’s broader claims about the human-likeness of learned representations are based on the full suite of evaluations described above and in the paper.
>
> **How was the CLIP model trained?**
>
> We did not use the pretrained weights of the original CLIP models. The visual features were precomputed from a vision transformer pretrained by unsupervised visual learning algorithms (DINO) on unlabeled ImageNet images. This is discussed in the middle of page 5, beginning at “Visual encoders and image features...” This selection of vision encoder is also explored in Section 4.6 and Figure 5. Thus, in this paper, we use “CLIP” to refer to the language and image contrastive training objective, and call models we trained using this objective function the “CLIP models.” As suggested by the reviewer, we have updated our paper to make this reference clearer (see page 5, we’re currently considering labeling them “CLI” models for “Contrastive Language/Image”).
>
> **Why these model architectures?**
>
> The CLIP, GIT, and Flamingo models represent very different multi-modality learning objective functions. As we draw conclusions from these different models, our results are more likely to generalize to other learning algorithms. For instance, the architecture proposed by Kiros et al. in 2014 utilized a structure-content neural language model module. This approach, similar to Flamingo, also implemented multi-modal interaction through a cross-modality-modulation technique. While we cannot test every possible architecture in the literature, we might expect similar findings based on structural similarity to Flamingo.
>
> Finally, we thank the reviewer for pointing out the typo (LM instead of LLM ) in our paper. We have fixed it. We also appreciate the reviewer’s suggestion about the line style, which we will actively think about to have a better solution.

---

> > ### Comment · Reviewer_QLPJ · 2023-11-14
> >
> > Thank you for the quick response! I'm now comfortable with your cognitive plausibility argument, and the point is still not perfectly clear to me in the paper---I noticed that you added "it is important to note that they are distinct from the pretrained CLIP
> > models developed by Radford et al. (2021)", and it'd be good (if I understood correctly) to note that you are using the same model architecture, adapting pretrained visual features, and completely training the language-module weights from scratch.
> >
> > On the other hand, I'm still unsure about the conclusion and, therefore, the contribution of this paper. While I agree with the authors that the models decently represent three vision-language modeling approaches, they are just a few among infinite models---it's hard to tell to what extent the conclusion holds for all vision-language models. Similar conclusions (i.e., visual grounding doesn't help much on pure-linguistics tasks) can also be found in various papers (e.g., Tan and Bansal (2020) mentioned by other reviewers), although this is not their major focus.

---

> > > ### Author Response · Authors · 2023-11-15
> > > **Response**
> > >
> > > We appreciate the quick response! About the CLIP models, you are right that we adapted pretrained visual features and completely trained the language-module weights from scratch. Our language module is not exactly the same as the pretrained CLIP model, as it has fewer layers (we have shown that this number of layers does not influence the result, see Appendix Fig. 8). We have updated the paper to add this point.
> > >
> > > As for the conclusion and contribution of this paper, we justify our selection of the models and the generalizability of our findings in the following two points.
> > > * One of our aims is to understand language acquisition in humans, and to help interpret the claims of existing studies that attempt to link acquisition and representation of language in humans and ML models. As a consequence, we are particularly interested in models that have demonstrated their abilities to capture dynamics in human brains. CLIP models have been shown to be accurate in predicting brain responses in many papers [1, 2, 3]. The GIT model has also been used to predict the caption of images seen by humans from their neuronal responses [4].
> > >
> > > * These three models represent very diverse methodologies for merging visual and language information. They have shown strong performance in real-world tasks, and exemplify what we consider to be the three predominant strategies for learning joint vision-and-language representations. Therefore, other multi-modality learning methods potentially share some architecture design with one of these three models, which makes our findings more likely to generalize to these other methods.
> > >
> > > In addition, we want to emphasize that our conclusion is not just that visual grounding only helps word learning in low-data regimes. We have also shown that simply applying the existing algorithms in small-sized datasets cannot effectively leverage the visual information. We have further identified that one of the key reasons for this inefficiency is the competence between linguistic distributional and visual information. Indeed, we find removing the linguistic distributional information yields a more efficient model in word learning in low-data regimes, which indicates that better resolving this competence is a promising future direction.
> > >
> > > Finally, we want to better distinguish the contributions in this paper from other papers. The studies mentioned by other reviewers (including the work by Tan and Bansal, 2020) either rely on existing models pretrained on billions of words or require large multi-modal training datasets. In contrast, our paper assesses the performance of models trained on smaller datasets, using word learning benchmarks. Our goal is twofold: firstly, to investigate various factors affecting the learning efficiency of language models, and secondly, to deepen our understanding of human language acquisition processes through these models. These objectives distinctly set our work apart from the studies mentioned by the reviewers. Finally, our approach contributes to the ongoing efforts in training language models by demonstrating how reducing data requirements, inspired by language acquisition in children, can be effective.
> > >
> > > [1] Liu, Yulong, et al. "BrainCLIP: Bridging Brain and Visual-Linguistic Representation via CLIP for Generic Natural Visual Stimulus Decoding from fMRI." arXiv preprint arXiv:2302.12971 (2023).
> > >
> > > [2] Wang, Aria Y., et al. "Incorporating natural language into vision models improves prediction and understanding of higher visual cortex." BioRxiv (2022): 2022-09.
> > >
> > > [3] Lu, Haoyu, et al. "Multimodal foundation models are better simulators of the human brain." arXiv preprint arXiv:2208.08263 (2022).
> > >
> > > [4] Ferrante, Matteo, et al. "Brain Captioning: Decoding human brain activity into images and text." arXiv preprint arXiv:2305.11560 (2023).

---

> > > > ### Comment · Reviewer_QLPJ · 2023-11-16
> > > >
> > > > Thanks for the justification of your model selection and further explanation! I really appreciate the active response from the authors.
> > > >
> > > > > Our goal is twofold: firstly, to investigate various factors affecting the learning efficiency of language models, and secondly, to deepen our understanding of human language acquisition processes through these models.
> > > >
> > > > I can't fully agree on either of the claimed contributions, although (I think) I got the ideas from the authors -- I apologize in advance if I misunderstood anything, and I'm happy to discuss further.
> > > > - First contribution: the primary goal of these language models is not learning word meanings, and it's probably more appropriate to say that these models are designed for learning visually grounded meanings for sentences. While word meanings can indeed be argued as the foundation, I would expect more analysis on the original evaluation metrics (e.g., sentence generation/retrieval) if model learning efficiency is claimed as a main contribution.
> > > > - Second contribution: I'm not sure if I see how this could advance our understanding of human language acquisition. Shouldn't we study human language acquisition through human experiments? Or are we assuming that these models are good enough to represent humans so that studying these models would necessarily lead to a better understanding of human brains?
> > > >
> > > > **Re. model selection**: [1-4] indeed shows positive evidence for the selected models on predicting human brain responses. However, I'm still not fully convinced about the model section: the main claims of [1-4] are showing positive results on certain models; in contrast, this paper aims to draw general conclusions for vision-language models, and the goal naturally necessitates a more comprehensive set of models.
> > > >
> > > > **Re. vision and language competence leads to inefficiency**: I'm sorry, but I didn't view this point as a major contribution of this paper as I thought this was quite intuitive from a machine learning perspective. Your evaluation is mostly on content words, and function words, or less visually groundable ones, could be viewed as noise. For Visual+Word model training, function words (noise) from the sentences are removed, and the model can focus more on the content words. It is, of course, nice to verify this, but I wouldn't necessarily be surprised by the results that Visual+Word outperforms Visual+Lang. in many settings, especially in low-data regimes where Visual+Lang. is not exposed to enough data to learn reasonably well.
> > > >
> > > > Still, I find the topic of this paper quite interesting, but the approach and conclusions need more justification.

---

> > > > > ### Author Response · Authors · 2023-11-19
> > > > > **Response (1/2)**
> > > > >
> > > > > Thanks for the quick response and active discussions!
> > > > >
> > > > > **Vision and language competence leads to inefficiency.**
> > > > >
> > > > > Your comment argues that improvement in the Visual + Word models is “intuitive from a machine learning perspective *because*  Visual + Word models did not learn from the “noise” words, as the word labels are only from the words of interest. To address this issue, we train new Visual + Word models on the *entire* vocabulary (without the word-of-interest limit). This means every word in a caption will be used as a separate label for its corresponding image. The results show that the performance of the Visual + Word (CLIP) models remains mostly the same. Moreover, we also train new Visual + Language (CLIP) models with captions only keeping the words of interest. We find that although this yields a slight improvement in the performance of Visual + Language models, they still significantly underperform both the Visual + Word (CLIP) and Language-Only models. Finally, we train new Language-Only models with captions only keeping the words of interest. These models yield slightly worse results than the original Language-Only models, which is inconsistent with the intuition that removing the “noise” words should yield better performance. All these results have been added to the paper in Appendix Fig. 17. Taken together, they clearly show that the improvement of Visual + Word models compared to the Visual + Language models and the benefits of visual grounding in low-data regimes cannot be explained by only having the word of interest.
> > > > >
> > > > > The comment also argues that Visual + Language models are “not exposed to enough data to learn reasonably well.” We argue that the Visual + Word models receive the same amount of information as the Visual + Language models. In fact, these two sets of models are trained using the same network architecture and learning algorithms, and the only difference between them is whether the linguistic distributional information is present or purposely removed during training. Therefore, we do not think Visual + Language models receive less data than Visual + Word models. The improvement of Visual + Word models over Visual + Language models is better explained by the competence between visual and linguistic distributional information.
> > > > >
> > > > > **Advancing our understanding of human language acquisition.**
> > > > >
> > > > > In response to the question about how our findings can advance the understanding of human language acquisition: In general, one important method to understand the language acquisition process is to construct **computational models** that predict children’s language acquisition behaviors when these models receive the same input children do. The models we are exploring in this work are all plausible **candidate models** for part of this process regarding how visual information is used by children in acquiring word meanings. In fact, the strong performance of Visual + Word (CLIP) models in low-data regimes provides a very compelling hypothesis about how children acquire language: they may very well also learn like Visual + Word models, meaning attempting to map each visual input to single words that are “loosely” tied to the input. We want to emphasize that what these models have achieved is not trivial at all. The Visual + Word models without the word-of-interest limit are asked to map every single word in the caption to the paired image, which includes many words that should not be considered related to the visual feature. This is a highly noisy learning environment, which is similar to the significant amount of noise in the learning signals children perceive. Still, they can efficiently acquire some aspects of word meanings on small datasets and are the most efficient method in some benchmarks, like the word-relatedness benchmark at low-data regimes. We believe these results indicate that this single-word-visual-mapping strategy is what children may employ during their early learning processes. We plan to further test how we can use the learning signatures of children to better validate this hypothesis in future work.

---

> > > > > > ### Author Response · Authors · 2023-11-19
> > > > > > **Response (2/2)**
> > > > > >
> > > > > > **Evaluating word learning instead of sentence generation/retrieval.**
> > > > > >
> > > > > > The comment also suggests that since these algorithms, like CLIP, GIT, and Flamingo, were developed for other tasks, like sentence generation and retrieval, the claim of efficiency should be based on evaluating these tasks, instead of word learning tasks. But the entire reason these models have been so successful (and the reason they’re starting to be taken seriously from a cognitive modeling standpoint), is that they perform well on more than just the tasks they were trained on. For example, the language modules in the pretrained CLIP models have proven useful in word-relatedness benchmarks [1]. On the other hand, the word-meaning tasks we have evaluated these models on are important tasks to test how part of word-level and sentence-level linguistic information is encoded in the intermediate layers of these models. This is crucial for understanding the internal representations of these models, and evaluating the learning efficiency of the models on these tasks is, therefore, critical to providing more details of the learning progress of these models beyond just the tasks they are trained on.
> > > > > >
> > > > > > **Model selection.**
> > > > > >
> > > > > > We want to argue that our current selection of architectures is already enough to support the key claims in this paper. Importantly, we want to emphasize that our results on Visual + Word (CLIP) models have clearly shown that visual grounding indeed helps word learning in low-data regimes, as these models outperform the Language-Only models on several important word-learning benchmarks in low-data regimes. As we have mentioned in the earlier part of this response, this key claim is not just an intuitive result from a machine-learning perspective. Simply training the existing algorithms on visually grounded datasets yields worse results (Visual + Language models) than both Visual + Word and Language-Only models. We further present analyses indicating that better merging visual and linguistic distributional information may lead to stronger models. Through our selection of three distinctive algorithms, we even go beyond just supporting the key claim that “visual-grounding helps word learning in low-data regimes,” and also show that the CLIP training method is more effective in leveraging visual information (when linguistic distributional information is absent) than the other models. Since these algorithms are both widely used and high-performing, we expect our findings to benefit the wide research community in exposing how visual and linguistic distributional information interacts during learning. Finally, the benchmarks we have used or developed in this paper will be released, which will encourage the submissions from other researchers to test more models. We will also actively develop and test more algorithms to further explore how visual information, or information from other modalities, can help improve the language acquisition efficiency in models.
> > > > > >
> > > > > > [1] Wolfe, Robert, and Aylin Caliskan. "Contrastive Visual Semantic Pretraining Magnifies the Semantics of Natural Language Representations." arXiv preprint arXiv:2203.07511 (2022).

---

### Author Response · Authors · 2023-11-14
**General response**

We thank the reviewers for their hard work in providing feedback on this work. We have replied to each reviewer separately and updated the paper to include more clarifications and explanations.